# Anthropometric and strength characteristics of adolescent golfers with low and high handicaps

Yaping Cao, Ju Li, Zhongcheng Li, Jian Lang *

College of P.E. and Sprorts, Beijing Normal University, Beijing, China

* langjian@bnu.edu.cn

## Abstract

### Background

The athletic performance of adolescent golfers is influenced by various factors, among which strength qualities and anthropometric characteristics are key. However, current research on these aspects among adolescent golfers with different handicaps remains limited.

### Objective

This study aimed to examine anthropometric and strength characteristics in adolescent golfers of differing handicaps and to explore their relationship with golf performance (handicap).

### Methods

This cross-sectional study recruited 40 adolescent golfers (25 males, 15 females) via convenience sampling, divided into low (n = 20) and high (n = 20) handicap groups. Sample size was determined by a priori power analysis. Anthropometric measures (height, shoulder width, hip, thigh, calf circumferences) and standardized strength tests (grip strength, medicine ball throws, standing long jump, countermovement jump) were assessed. Group differences were analyzed via independent t-tests, and correlations between handicap and strength metrics were analyzed using Pearson's correlation (p < 0.05).

### Results

Golfers with low handicaps demonstrated significantly greater shoulder width (p = 0.033), hip, thigh, and calf circumferences (p < 0.01), and performed better in all strength tests (p < 0.01). Pearson correlation analysis indicated significant negative correlations between handicap and multiple strength metrics (r ranging from -0.324 to -0.556, p < 0.05). Multiple regression analysis showed that strength variables explained approximately 60% of handicap variance. No gender comparisons were conducted.

**Data availability statement:** We would like to clarify that although we initially declared that all relevant data are fully available without restriction, after further consultation with our institutional ethics committee, we have been advised that, due to privacy and confidentiality considerations involving adolescent participants (aged 12–16), sharing the complete raw dataset publicly is not appropriate. Therefore, the data underlying the findings of this study are available upon reasonable request for qualified researchers who meet the criteria for access to confidential data. Data Access Contact (primary): Beijing Normal University – College of Physical Education and Sports Ethics Committee Secretariat Email: ethics.edu@outlook.com Additional contact (corresponding author): Dr. Jian Lang Email: langjian@bnu.edu.cn The data are securely stored under the supervision of the ethics committee and will be accessible for at least 10 years upon request, in accordance with ethical and institutional standards.

**Funding:** The author(s) received no specific funding for this work.

**Competing interests:** The authors have declared that no competing interests exist.

## Conclusion

Anthropometric advantages and higher strength/explosive power are associated with better performance (lower handicap) in adolescent golfers. It is recommended that training programs for adolescent golfers emphasize strength and explosive power development to improve competitive performance.

## 1 Introduction

Golf is a highly technical sport that requires precise coordination, muscular strength, and optimal anthropometric characteristics to achieve competitive success. Numerous studies have explored the relationship between physical fitness factors such as muscular strength, explosive power, and golf performance, consistently highlighting their positive influence on key performance indicators like clubhead speed and driving distance [1–3]. For instance, strength training interventions have demonstrated effectiveness in significantly enhancing clubhead speed and driving distances, contributing to improved competitive results in golfers across different age categories [4–6].

Anthropometric characteristics also play a pivotal role in golfers' performance by affecting swing stability, rotational power, and biomechanical efficiency. Specifically, physical features such as broader shoulder width provide enhanced stability and leverage, which directly influence swing mechanics, enabling golfers to generate higher clubhead velocities [7]. Hip and thigh circumferences correlate positively with musculature size and strength, critical for generating power and maintaining stability throughout the golf swing [8,9]. Furthermore, greater muscle mass in lower limbs, as reflected by thigh and calf circumferences, is associated with improved ground force utilization and better balance, thereby enhancing overall swing performance and consistency [10,11].

Previous literature primarily emphasizes strength and explosive power as primary determinants of golf performance [12,13], while studies systematically investigating anthropometric factors are relatively limited. However, existing research does indicate that anthropometric features significantly influence biomechanical and physiological capabilities in golfers. Keogh et al. (2009), for instance, observed clear relationships between anthropometric variables—such as limb circumference, shoulder width, and body proportions—and clubhead speed in golfers. Similarly, Newman et al. (2016) reported that hip morphology and musculature size directly impact golf swing mechanics and performance outcomes among professional golfers.

Although the general relationship between muscular strength, anthropometric characteristics, and motor performance is well-established in sports sciences [14,15], the unique rationale for focusing specifically on adolescent golfers merits further clarification. Adolescence is a critical developmental period characterized by significant changes in muscle mass, body composition, and motor coordination, all of which substantially influence performance outcomes [10,16]. Investigating these characteristics during adolescence provides valuable insights for targeted training interventions,

early talent identification, and long-term athlete development, significantly enhancing the potential for future competitive success in golf [1,2].

Anthropometric and physical fitness assessments have been widely adopted as essential criteria for talent identification and athlete selection across various adolescent sports due to their predictive value regarding future athletic performance [17,18]. Specifically, evaluating these attributes during adolescence offers essential educational insights for coaches and trainers to design individualized training programs and development strategies tailored explicitly for optimizing young athletes' potential. Additionally, it enables early identification of high-potential talents, which is crucial for systematically nurturing and enhancing competitive abilities throughout their athletic career.

During adolescence, improvements in muscular strength significantly enhance an athlete's capability to produce higher clubhead velocities, maintain swing consistency, and optimize biomechanical efficiency [1,12]. Concurrently, rapid growth in anthropometric dimensions, notably limb circumference and muscle mass, profoundly impacts biomechanical performance by enhancing joint stability, force generation capabilities, and overall motor control [8,10]. Consequently, comprehensive knowledge about adolescent strength and anthropometric characteristics becomes essential in understanding and subsequently enhancing sports performance in junior golf athletes.

However, despite the recognized importance of anthropometric and muscular strength characteristics, few studies have specifically compared these variables between low-and high-handicap adolescent golfers. Given that adolescence represents a crucial developmental period marked by rapid growth in muscle mass and physical capabilities, systematically investigating these differences is necessary. Identifying the specific physical attributes that differentiate golfers at different skill levels during adolescence can guide targeted training programs, inform talent identification, and optimize athlete development strategies.

Given this background, the current study aims to systematically investigate the anthropometric and muscular strength differences between adolescent golfers with low and high handicaps. Clarifying these differences will help inform targeted training and coaching practices, enhancing performance and facilitating optimal talent identification and development. We hypothesize that adolescent golfers in the low handicap group will exhibit superior anthropometric and strength characteristics compared to their high-handicap counterparts.

## 2 Materials and methods

### 2.1 Participants

This study employed a cross-sectional design, with participants recruited through convenience sampling from local adolescent golf programs between July 16 and September 16, 2022. A total of 40 adolescent golfers (25 males, 15 females) aged between 12 and 16 years were recruited, with an average age of 13.98 ± 1.19 years. All participants were right-handed golfers, training at least twice a week for 2 hours each session, and had at least two years of golf training experience. Participants were divided into low handicap and high handicap groups based on their handicap levels, with 20 participants in each group. The low handicap group included 12 males and 8 females, and the high handicap group had 13 males and 7 females, achieving a comparable gender distribution. No additional criteria aside from handicap were used for grouping. The average handicap of the low group was 6.30 ± 2.11, while the high group's average was 20.00 ± 1.41.

The sample size for this study was estimated through an a priori statistical power analysis using G*Power 3.1 software [19]. Based on previous literature examining similar populations [1], we anticipated a moderate-to-large effect size (Cohen's d ≈ 0.8) for anthropometric and strength comparisons between low and high handicap adolescent golfers. To achieve a power (1-β) of 0.80, with an alpha (α) of 0.05 for detecting significant differences using independent t-tests, a minimum sample size of approximately 34 participants (17 per group) was required. Accounting for potential dropout or measurement errors, a final sample of 40 participants (20 per group) was determined to ensure sufficient statistical power and reliable conclusions.

In this study, gender distribution was carefully considered when grouping participants. Specifically, the low handicap group included 12 males and 8 females, while the high handicap group consisted of 13 males and 7 females. The gender proportions were therefore comparable between groups, which minimized potential gender-based biases. However, no separate gender-based comparisons or gender-specific correlations were conducted due to the limited subgroup sizes. Future research with larger sample sizes may explore gender-specific differences more explicitly.

Participants refrained from any intense physical activity within 48 hours before the test and had no history of muscle, skeletal injuries, or diseases in the past six months. To ensure the reliability of the test data, all testers completed training and guidance for the entire testing process before the official test. All tests were performed on the same day, following identical standards and sequence. Informed consent, in writing, was obtained from all participants and their guardians prior to the commencement of the study. The study followed the Declaration of Helsinki guidelines and received approval from the Institutional Review Board of the College of P.E. and Sports at Beijing Normal University.

## 2.2 Procedures

All assessments were carried out in the laboratory using identical testing equipment and measurement protocols. The procedures were performed by two systematically trained testers who adhered to the same standards and sequence. Before testing, all Participants were informed of the detailed testing procedures and provided with standardized verbal prompts. All Participants performed a 10-minute standardized dynamic activation and thorough warm-up before the tests. Height (cm) and weight (kg) were measured before the strength tests. All Participants self-reported their latest golf handicap.

### 2.2.1 Anthropometry.

Trained and experienced testers conducted anthropometric measurements following the protocol set by the International Society for the Advancement of Kinanthropometry (ISAK). Participants were barefoot and wore light sports shorts and shirts during the measurements.

Height (cm): Recorded to the nearest 0.1 cm using a portable stadiometer (YL-65S, Yagami, Nagoya, Japan).

Weight (kg): Determined to the nearest 0.1 kg with a calibrated scale (Big Pro, XiaoMi, Beijing, China) [20].

Arm Span (cm): Measured as the distance between the tips of the middle fingers of both hands extended parallel to the ground, using a standard measuring tape (DL9810, Deli, Beijing, China) to the nearest cm.

Shoulder Width (cm): Measured as the distance between the acromia of both shoulders, to the nearest cm.

Chest Circumference (cm): Measured from the highest point of the left side of the chest (sternal edge) around the chest to the starting point, to the nearest cm.

Upper Arm Circumference (cm): Measured at the midpoint between the acromion and the olecranon with the arm relaxed, to the nearest cm.

Left and Right Thigh Circumference (cm): Measured at the level 5 cm proximal to the upper edge of the patella while the subject stands with the thigh and calf axes perpendicular, to the nearest cm [21].

Left and Right Calf Circumference (cm): Measured at the thickest part of the calf while the subject stands with the thigh and calf axes perpendicular, to the nearest cm [21].

Hip Circumference (cm): Measured at the widest part of the hips with the subject standing with feet together, to the nearest cm.

Leg Length (cm): Measured from the ischial tuberosity to the top of the medial malleolus with the subject seated and legs hanging vertically, to the nearest cm.

These specific sites were chosen for anthropometric measurements because the muscle cross-sectional area at these locations is positively correlated with muscle strength and explosive power [22].

### Sources of Anthropometric Tests

Anthropometric measurements including arm circumference,chest circumference, and shoulder width were performed according to standardized anthropometric protocols established by the International Society for the Advancement of

Kinanthropometry (ISAK) guidelines [23]. These standardized methods have been extensively validated and recommended for adolescent athletes.

### 2.2.2 Strength testing.

Strength tests were conducted using standardized equipment and methods to measure the following indicators:

**Maximum Grip Strength** (kg):

Measurements were taken using a portable Smedley III T-18A dynamometer (Takei, Tokyo, Japan). The device has a testing range from 0 to 100 kg, increments of 0.5 kg, and an accuracy of ±2 kg. Participants stood naturally with straight arms, keeping their arms fully extended downward along their sides, without swinging or bending the elbow, performing a maximal voluntary contraction by squeezing the dynamometer as forcefully as possible for 3 seconds using their fingers [24]. Each subject repeated the test three times for each hand with a 60-second rest interval between tests. The maximum grip strength of the left and right hands was recorded.

Grip strength is critical for golfers as it influences club control, swing stability, and power transfer. Stronger grip strength helps golfers maintain consistent clubface alignment and reduce swing variability.

**Forward Medicine Ball Throw** (m):

Participants stood behind the throwing line, feet parallel and shoulder-width apart, holding a 2 kg medicine ball with both hands. They threw the ball forward from above their head with maximum effort without moving their feet or crossing the line [25]. The distance from the line to where the ball landed was measured. Each participant performed three trials with a 60-second rest interval, and the longest throw was recorded for analysis. Participants began with the ball held overhead, arms fully extended, and threw the ball forward using a rapid, coordinated motion involving the upper limbs, core, and minimal torso rotation, without stepping forward or losing balance.

This test measures explosive upper limb and core strength, reflecting the golfer's ability to generate high initial clubhead speed and control.

**Backward Medicine Ball Throw** (m):

Participants held a 2 kg medicine ball with both hands, standing behind the throwing line and facing away from the throwing direction, with feet parallel and shoulder-width apart. Without crossing the line, they threw the ball backward from their chest with maximum effort. The distance from the throwing line to the point where the ball landed was measured. Each subject repeated the test three times with a 60-second rest interval between tests, and the longest throw was recorded for analysis. Participants held the ball at chest height, elbows bent slightly outward, and threw it backward explosively without moving their feet or stepping over the line.

This test assesses explosive strength of the back and core muscles, key components for the rotational force and power production during a golf swing.

**Left and Right Side Medicine Ball Throw** (m):

Side medicine ball throws are commonly used to measure the strength and explosive power of golfers and have been proven to exhibit high consistency upon repeated testing [26]. Participants used a 2 kg medicine ball, standing sideways with feet shoulder-width apart. They threw the ball from the side of their body to the front side using a pivoting motion. The distance from the line to where the ball first landed was measured. Each side was tested three times with a 60-second rest between trials, and the longest throw was used for analysis.

During the lateral throws, participants kept their feet in contact with the ground, (allowing the back heel to lift and move similar to a golf swing). They threw the medicine ball from the side-back to the side-front. The throw was considered successful if the ball landed within a 1.5-meter-wide area. If not successful, the test was repeated until it was. The distance from the throwing line to the ball's first landing point was measured to the nearest 5 cm. The farthest distance was used for data analysis. Participants stood perpendicular to the throwing direction, feet shoulder-width apart, performing a pivoting rotation of hips and torso similar to a golf swing while throwing the medicine ball laterally with maximal effort. The test was repeated on both sides.

Lateral throws specifically evaluate rotational power and core strength essential for efficient swing mechanics, clubhead velocity, and shot distance in golf.

**Validity, Reliability and Sources of Strength Tests**

The maximum grip strength test employed a Smedley III T-18A hand dynamometer, a valid and reliable device widely used in adolescent populations, showing excellent test-retest reliability (ICC range: 0.90–0.96) and good validity [27]. The medicine ball throw tests (forward, backward, right, and left throws) used in this study are commonly adopted field assessments in sports science research to evaluate upper limb and core explosive power. Although specific ICC values for these directional throws were not located, previous studies generally support the practical validity and test-retest reliability of medicine ball throw protocols as effective performance tests among youth athletes [26,28,29].

**Standing Long Jump** (m):

The standing long jump primarily measures lower body explosive power and is regarded as an effective and dependable method for evaluating lower limb strength and explosiveness [30]. Participants stood behind the take-off line with feet naturally apart and parallel, then jumped forward with maximum effort. The distance from the take-off line to the heel of the landing foot was measured. Each participant completed the test three times with a 60-second rest between attempts, and the longest distance was recorded for analysis. Participants stood with feet shoulder-width apart, knees slightly bent, swinging their arms forward while jumping horizontally with maximal effort, landing simultaneously on both feet.

Standing long jump reflects lower-body explosive power, crucial for generating ground reaction force, swing stability, and efficient kinetic energy transfer from the lower body to the upper limbs during the golf swing.

**Countermovement Jump Test** (cm):

The countermovement jump is frequently used to assess lower body explosive power and is known for its strong validity and consistency upon repeated testing [31]. Jump height was measured using an optical system (Optojump, Microgate, Bolzano-Bozensh, Italy). Participants selected their preferred jump depth and arm swing before jumping as high as possible. Each participant completed the test three times with a 60-second rest between trials, and the best result was used for analysis. Participants started in an upright position, performed a rapid downward movement by flexing hips and knees, then immediately jumped vertically as high as possible, arms freely assisting the movement.

This vertical jump test assesses lower limb explosive power and neuromuscular coordination, both key factors for dynamic stability and effective power generation during golf swings.

**Validity, Reliability and Sources of Strength Tests**

The maximum grip strength test employed a Smedley III T-18A hand dynamometer, a valid and reliable device widely used in adolescent populations, showing excellent test-retest reliability (ICC range: 0.90–0.96) and good validity [27].

The medicine ball throw tests (forward, backward, right, and left throws) used in this study are commonly adopted field assessments in sports science research to evaluate upper limb and core explosive power. Although specific ICC values for these directional throws were not located, previous studies generally support the practical validity and test-retest reliability of medicine ball throw protocols as effective performance tests among youth athletes [26,28,29].

### 2.3 Statistical analysis

All data were analyzed using SPSS 26.0 statistical software. Descriptive statistics were initially performed to compute the mean and standard deviation for each metric. The Shapiro-Wilk test was used to assess the normality of all data, determining if they followed a normal distribution.

For normally distributed data, differences in anthropometric and strength characteristics between the low and high handicap groups were analyzed using an independent samples t-tests were used to compare groups ($\alpha = 0.05$). For variables not meeting normality, a Mann-Whitney U test was planned. Pearson's correlation analysis was used to examine relationships between each metric and handicap ($\alpha = 0.05$). Additionally, a multiple linear regression analysis was conducted to identify which anthropometric and strength variables best predict handicap. The model's coefficient of

determination (R²) was reported to assess predictive power. The Shapiro-Wilk test confirmed all variables were normally distributed ($p > 0.05$), so parametric tests were applied and the Mann-Whitney U test was not required.

All statistical analyses were performed at a two-sided significance level of 0.05. Results were reported as mean ± standard deviation (M ± SD).

## 3 Results

### 3.1 Descriptive statistics

This study included a total of 40 adolescent golfers, with 20 golfers in each of the low and high handicap groups. No significant differences were observed between the two groups in basic characteristics such as age, height, and weight ($p > 0.05$), as presented in Table 1. However, the low handicap group exhibited significantly superior anthropometric measurements in shoulder width, hip circumference, as well as left and right thigh and calf circumferences ($p < 0.05$), detailed in Table 1

Statistical analysis showed that all variables conformed to normal distribution ($p > 0.05$, Shapiro-Wilk test); therefore, independent sample t-tests were applied for all variables. The Mann-Whitney U test was originally planned but not utilized, as the data met parametric assumptions.

### 3.2 Strength test results

The low handicap group showed significant advantages in all strength test items. The maximum grip strength of the left hand and right hand were 26.94 ± 6.56 kg and 29.04 ± 7.74 kg, respectively, while the high handicap group had 21.38 ± 8.75 kg and 23.61 ± 8.32 kg, ($p < 0.05$). In the forward, backward and left-side medicine ball throw tests, the low handicap group scored 8.41 ± 2.67 m, 9.31 ± 3.39 m, and 8.63 ± 2.75 m, respectively, significantly better than the high handicap group's 5.63 ± 1.96 m, 6.12 ± 2.95 m, and 5.54 ± 2.32 m ($p < 0.01$). Additionally, the low handicap group outperformed the high handicap group in the standing long jump and countermovement jump tests ($p < 0.01$), as shown in Table 2.

### 3.3 Correlation analysis

Pearson correlation analysis revealed significant negative correlations between handicap and several strength test metrics, including maximum grip strength of the left hand ($r = -0.556$, $p < 0.01$), forward medicine ball throw ($r = -0.506$, $p < 0.01$), backward medicine ball throw ($r = -0.443$, $p < 0.01$), left-side medicine ball throw ($r = -0.524$, $p < 0.01$), standing long jump ($r = -0.556$, $p < 0.01$), and countermovement jump ($r = -0.528$, $p < 0.01$). This indicates that strength and explosive power significantly influence the competitive level of adolescent golfers. Table 3 details the correlations between each test

Table 1. Basic characteristics and anthropometric measurements of adolescent golfers with low and high Handicaps.

| Variable | Low Handicap Group (n=20) | High Handicap Group (n=20) | t-value | p-value |
|---|---|---|---|---|
| Age (years) | 13.98 ± 1.19 | 13.85 ± 1.14 | 0.350 | 0.729 |
| Height (cm) | 169.25 ± 8.81 | 162.85 ± 12.33 | 1.888 | 0.067 |
| Weight (kg) | 59.70 ± 14.18 | 57.81 ± 29.34 | 0.259 | 0.797 |
| Shoulder Width (cm) | 42.05 ± 3.71 | 39.30 ± 4.16 | 2.209 | 0.033* |
| Hip Circumference (cm) | 94.25 ± 9.98 | 86.90 ± 9.49 | 2.387 | 0.022* |
| Left Thigh Circumference (cm) | 51.44 ± 6.61 | 44.42 ± 8.05 | 3.015 | 0.005** |
| Right Thigh Circumference (cm) | 51.63 ± 6.25 | 45.30 ± 7.36 | 2.934 | 0.006** |
| Left Calf Circumference (cm) | 36.42 ± 3.38 | 32.97 ± 4.54 | 2.724 | 0.010** |
| Right Calf Circumference (cm) | 36.64 ± 3.16 | 33.45 ± 4.10 | 2.757 | 0.009** |

*$p < 0.05$ **$p < 0.01$.

$P < 0.05$ denotes significant differences, $P < 0.01$ represents highly significant differences.

**Table 2. Strength test results of adolescent golfers with low and high handicaps.**

| Variable | Low Handicap Group (n = 20) | High Handicap Group (n = 20) | t-value | p-value |
|---|---|---|---|---|
| Maximum Grip Strength (Left Hand) (kg) | 26.94 ± 6.56 | 21.38 ± 8.75 | 2.272 | 0.029* |
| Maximum Grip Strength (Right Hand) (kg) | 29.04 ± 7.74 | 23.61 ± 8.32 | 2.134 | 0.039* |
| Forward Medicine Ball Throw (m) | 8.41 ± 2.67 | 5.63 ± 1.96 | 3.760 | 0.001** |
| Backward Medicine Ball Throw (m) | 9.31 ± 3.39 | 6.12 ± 2.95 | 3.178 | 0.003** |
| Left-Side Medicine Ball Throw (m) | 8.63 ± 2.75 | 5.54 ± 2.32 | 3.848 | 0.000** |
| Standing Long Jump (m) | 1.95 ± 0.20 | 1.61 ± 0.26 | 4.577 | 0.000** |
| Countermovement Jump (cm) | 35.25 ± 10.48 | 24.13 ± 6.85 | 3.971 | 0.000** |

*$p < 0.05$ **$p < 0.01$.

$P < 0.05$ denotes significant differences, $P < 0.01$ represents highly significant differences.

**Table 3. Correlation between strength test metrics and handicap.**

| Test Metric | Pearson (r) | p-value |
|---|---|---|
| Maximum Grip Strength (Left Hand) (kg) | -0.556 | 0.000** |
| Forward Medicine Ball Throw (m) | -0.506 | 0.000** |
| Backward Medicine Ball Throw (m) | -0.443 | 0.000** |
| Left-Side Medicine Ball Throw (m) | -0.524 | 0.000** |
| Standing Long Jump (m) | -0.556 | 0.000** |
| Countermovement Jump (cm) | -0.528 | 0.000** |

$p < 0.01$ indicates a significant negative correlation.

metric and handicap. No significant correlation was found between players' anthropometric measurements (e.g., shoulder width, limb circumferences) and handicap in the overall sample. Therefore, a correlation table for anthropometric variables is not included. The between-group differences in these measures are presented in Table 1.

This finding indicates that strength and explosive power significantly influence the competitive level of adolescent golfers. For example, the strongest observed correlation (between left-hand grip strength and handicap, $r = -0.556$) corresponds to $R^2 \approx 0.309$, meaning about 30.9% of the variance in handicap can be explained by this factor. This indicates that left-hand grip strength alone explains about 30.9% of the variability in handicap scores, demonstrating its substantial predictive influence on performance. Other strength-related variables showed moderate predictive power, with $R^2$ values ranging from approximately 10% to 31%, emphasizing the varying importance of individual strength metrics in explaining differences in golf performance among adolescents.

Furthermore, multiple regression analysis revealed that left-hand grip strength and standing long jump distance were significant independent predictors of handicap (both $p < 0.01$). The overall regression model explained approximately 60% of the variance in handicap ($R^2 \sim 0.60$), highlighting the substantial predictive power of these strength measures.

## 4 Discussion

This study aimed to investigate differences in anthropometric and strength characteristics between adolescent golfers with low and high handicaps, as well as their correlations with golf handicap. Results indicated that low-handicap adolescent golfers exhibited significantly superior anthropometric attributes, specifically greater shoulder width, hip circumference, thigh circumference, and calf circumference. Additionally, the low-handicap group outperformed the high-handicap group across multiple strength and power measures, including grip strength, forward, backward, and lateral medicine ball throws, standing long jump, and countermovement jump.

## 4.1 Anthropometric characteristics

Adolescent golfers with lower handicaps displayed notably better anthropometric profiles, including broader shoulders and greater hip, thigh, and calf circumferences. Broader shoulders provide enhanced swing stability and increased rotational torque, allowing golfers to generate higher clubhead speeds and driving distances [7,9]. Larger hip circumference typically indicates stronger gluteal muscles, crucial for maintaining stability and efficient power transfer during swings [8,11]. Additionally, greater thigh circumference reflects increased quadriceps and hamstrings strength, significantly enhancing lower-body stability and explosive power during the swing [10,32]. Similarly, larger calf circumference, indicative of stronger lower limb musculature, improves ground reaction force utilization, balance, and overall swing consistency [33,34].

Despite these observed anthropometric advantages in the low-handicap group, the current study found no significant correlations between specific anthropometric measures (e.g., height, arm span, lower limb length) and handicap across the whole sample. This suggests that while advantageous anthropometric features may facilitate superior performance, they alone are insufficient predictors of golf handicap. Previous studies have also noted similar findings, indicating that technical skills, muscular strength, and neuromuscular coordination may more directly impact performance outcomes than anthropometric characteristics alone [7,20]. Therefore, anthropometric assessments should be integrated into a broader training context focusing primarily on strength and technical skill development.

## 4.2 Strength and explosive power characteristics

Adolescent golfers in the low-handicap group significantly outperformed their high-handicap counterparts in all strength and explosive power assessments, including maximum grip strength, medicine ball throws (forward, backward, lateral), standing long jump, and countermovement jump. These results highlight clear advantages in upper limb strength, core power, and lower limb explosive capabilities, essential for achieving high clubhead speeds and optimal swing mechanics.

Grip strength specifically reflects upper limb muscular strength, directly enhancing club control, reducing vibration, and improving swing consistency and ball speed [35,36]. Medicine ball throw distances are closely related to core and rotational power, critical for generating high swing velocities and efficient biomechanical energy transfer [26,28]. The observed superior performances in standing long jump and countermovement jump tests indicate enhanced lower limb explosive power, essential for maximizing ground reaction forces and neuromuscular coordination during golf swings [12,37].

These findings align well with previous studies. Coughlan et al. (2020) and Lewis et al. (2016) similarly reported significant correlations between grip strength, medicine ball throw distances, lower-limb explosive power, and golfing performance. Such consistency underscores the predictive value of strength and power measures for performance outcomes in adolescent golfers.

**Correlation analysis.** To further substantiate the significance of strength-related metrics, correlation analyses were performed. Significant negative correlations were identified between handicap and several strength variables, notably left-hand grip strength ($r = -0.556$, $p < 0.01$), forward medicine ball throw ($r = -0.506$, $p < 0.01$), backward medicine ball throw ($r = -0.443$, $p < 0.01$), lateral medicine ball throw ($r = -0.524$, $p < 0.01$), standing long jump ($r = -0.556$, $p < 0.01$), and countermovement jump ($r = -0.528$, $p < 0.01$). Left-hand grip strength alone explained approximately 30.9% of the variance in handicap ($R^2 \approx 0.309$).

These findings emphasize the critical role that strength and explosive power play in differentiating performance levels among adolescent golfers and reinforce the value of targeted strength training interventions.

## 4.3 Educational and practical implications

Given these clear relationships between anthropometric characteristics, muscular strength, and golf performance, the educational and practical applications of the study are substantial. Coaches and physical educators should systematically

incorporate anthropometric evaluations and targeted strength training into youth golf programs. Specifically, training interventions focusing on grip strength enhancement, core rotational power (medicine ball throws), and lower limb explosive exercises (e.g., plyometric jumps) should be regularly integrated into adolescent golfers' training schedules.

Early implementation of structured strength and conditioning programs during adolescence—a critical developmental period characterized by rapid physical growth—can significantly optimize physical development, enhance technical skills, improve performance consistency, and reduce injury risks. Moreover, systematically assessing these characteristics can substantially improve talent identification and selection processes, enabling the development of tailored training programs that support long-term athlete progression and competitive success [2,16].

### 4.4 Limitations and future research directions

While this study highlights significant associations between anthropometric, strength characteristics, and golf performance, several limitations should be considered. First, the convenience sampling method and relatively small sample size limit the generalizability of these findings. Secondly, gender-specific comparisons were not performed due to limited subgroup sizes, an area recommended for future investigation. Finally, given the cross-sectional design, causal relationships cannot be established.

Future longitudinal studies with larger, diverse samples are recommended to explore causal effects and gender-specific differences comprehensively, ultimately enhancing training efficacy and athlete development strategies.

## 5 Conclusion

Adolescent golfers with low handicaps exhibit significantly superior anthropometric and strength characteristics compared to their high-handicap peers, demonstrating the crucial role these physical attributes play in golf performance. Targeted strength and explosive power training, systematically integrated into youth training programs, can optimize athletic development, improve competitive outcomes, and support effective talent identification. Further longitudinal research is warranted to deepen our understanding of these relationships and improve long-term athlete development strategies in adolescent golfers.

## Acknowledgments

Yaping Cao put forward the research idea, designed the research idea and framework, and wrote the manuscript. Yaping Cao, Jian Lang and Zhongcheng Li, participated in the experimental test and analysis. Ju Li carried out some mathematical statistical analysis. Each author contributed to the first draft and worked together on the final version.

## Author contributions

**Conceptualization:** YaPing Cao, Jian Lang.

**Data curation:** YaPing Cao, Ju Li.

**Formal analysis:** Ju Li, Zhongcheng Li.

**Investigation:** YaPing Cao.

**Methodology:** YaPing Cao, Ju Li, Zhongcheng Li, Jian Lang.

**Project administration:** Jian Lang.

**Resources:** Jian Lang.

**Supervision:** Zhongcheng Li, Jian Lang.

**Validation:** Ju Li.

**Writing – original draft:** YaPing Cao.

**Writing – review & editing:** YaPing Cao.

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
