## [Decision Letter · Decision Letter 0]

15 Dec 2024

PONE-D-24-29304Anthropometric and Strength Characteristics of Adolescent Golfers with Different HandicapsPLOS ONE

Dear Dr. Lang,

Thank you for submitting your manuscript to PLOS ONE. After careful consideration, we feel that it has merit but does not fully meet PLOS ONE’s publication criteria as it currently stands. Therefore, we invite you to submit a revised version of the manuscript that addresses the points raised during the review process.

 Dear authors: Please revise your manuscript according to the reviewers' comments, then re-submitted it for full consideration.

We look forward to receiving your revised manuscript.

Kind regards,

Rasool Abedanzadeh, Ph.D

Academic Editor

PLOS ONE

Journal Requirements:

2. We note that your Data Availability Statement is currently as follows: [All relevant data are within the manuscript and its Supporting Information files.] Please confirm at this time whether or not your submission contains all raw data required to replicate the results of your study. Authors must share the “minimal data set” for their submission. PLOS defines the minimal data set to consist of the data required to replicate all study findings reported in the article, as well as related metadata and methods (https://journals.plos.org/plosone/s/data-availability#loc-minimal-data-set-definition). For example, authors should submit the following data: - The values behind the means, standard deviations and other measures reported; - The values used to build graphs; - The points extracted from images for analysis. Authors do not need to submit their entire data set if only a portion of the data was used in the reported study. If your submission does not contain these data, please either upload them as Supporting Information files or deposit them to a stable, public repository and provide us with the relevant URLs, DOIs, or accession numbers. For a list of recommended repositories, please see https://journals.plos.org/plosone/s/recommended-repositories. If there are ethical or legal restrictions on sharing a de-identified data set, please explain them in detail (e.g., data contain potentially sensitive information, data are owned by a third-party organization, etc.) and who has imposed them (e.g., an ethics committee). Please also provide contact information for a data access committee, ethics committee, or other institutional body to which data requests may be sent. If data are owned by a third party, please indicate how others may request data access.

Reviewers' comments:

Reviewer's Responses to Questions

**Comments to the Author**

1. Is the manuscript technically sound, and do the data support the conclusions?

Reviewer #1: Yes

Reviewer #2: Yes

2. Has the statistical analysis been performed appropriately and rigorously? 

Reviewer #1: Yes

Reviewer #2: No

3. Have the authors made all data underlying the findings in their manuscript fully available?

Reviewer #1: No

Reviewer #2: Yes

4. Is the manuscript presented in an intelligible fashion and written in standard English?

Reviewer #1: Yes

Reviewer #2: Yes

5. Review Comments to the Author

Reviewer #1: You carried out the research by observing the principles of ethics in research. I hope that the mentioned reforms will be carried out and reviewed so that it can be used correctly by other researchers to conduct research in the future.

Reviewer #2: ________________________________________

Review of the Article: "Anthropometric and Strength Characteristics of Adolescent Golfers with Different Handicaps"

Dear Authors,

To enhance the quality of this article, I kindly suggest the following revisions:

Title:

The title should be revised to better reflect the study's objective.

Suggested Revision:

"Anthropometric and Strength Characteristics of Adolescent Golfers with Low and High Handicaps."

Abstract:

• Please specify the sampling method used and provide a clear explanation of how the sample size was determined.

• The exact p-value should be stated.

• If gender comparisons were made, please indicate this and clarify whether correlations were assessed based on gender.

• The conclusion section should be revised to focus on the implications of the findings rather than restating the results.

• It is advisable to use keywords that are distinct from those already included in the title.

• The abstract appears to be too lengthy. Consider shortening it to a maximum of 250 words.

Introduction:

• Ensure that all statements are appropriately cited with references.

• Provide more detailed information about the role of anthropometric characteristics in golfers' performance.

• The literature review largely focuses on the role of strength and power in performance. It is essential to also review studies that explore the significance of anthropometric features in this context.

• While the role of muscular strength and anthropometric characteristics on motor performance is well established, the unique rationale for this research needs further clarification.

• Since anthropometric and physical fitness (strength) assessments serve as entry criteria for most sports and are key to talent identification, please justify the necessity of examining these factors at the adolescent level. Provide educational reasoning for their inclusion in this study.

• Offer a more detailed explanation of how adolescent strength and anthropometric characteristics impact sports performance.

• Numerous studies have explored the importance of anthropometric features and strength in adolescent athletes. Drawing upon relevant studies will strengthen the introduction and provide a more solid foundation.

Research Method:

Participants:

• Please specify the sampling method and the process for estimating the sample size.

• Considering gender differences among adolescents, how were the groups balanced by gender?

• Were there any other criteria, aside from handicap, used for grouping participants? Please clarify.

Statistical Analysis:

• Given the multi-level nature of the research variables, please report the results of multiple regression analysis, if applicable, to determine which variables have the highest predictive power.

• When discussing statistical tests, simply mention the test used and justify its selection. Avoid additional explanations.

Results:

• Modify the titles of the tables to better align with the study’s objectives (i.e., “Golfers with Low and High Handicaps”).

• In Table 3, please report the R² value and provide an analysis based on this value.

• A correlation table for the anthropometric characteristics would be beneficial. Please explain why such a table was not included.

• It would be useful to present the results of multiple regression analysis to highlight the predictive power of the strength and anthropometric variables.

• In the methods section, it is stated that the Mann-Whitney U test was used, but this is not mentioned in the results section. Please ensure consistency.

Discussion:

• Given the length of the discussion, I suggest you begin with a concise summary of the findings related to anthropometric characteristics and strength analysis. Then, compare these findings with those of previous studies in this area, and conclude with a general summary.

• Given the relationship between limb volume and strength (a key factor in improving performance), I recommend further analysis of the results in terms of their educational applications.

• It might be better to present the correlation section towards the end of the strength discussion.

• Please include practical suggestions based on the findings, particularly from an educational perspective.

References:

• The font style in references 2, 5, 12, 26, 29, 39, 42, and 46 appears to be inconsistent. Please correct this formatting issue.

6. PLOS authors have the option to publish the peer review history of their article (what does this mean?). If published, this will include your full peer review and any attached files.

Reviewer #1: No

Reviewer #2: No

---

## [Author Response · Author response to Decision Letter 0]

30 Mar 2025

Dear Editor,

Thank you very much for your detailed feedback and continued guidance regarding the data availability requirements. We sincerely appreciate your support in promoting transparency and long-term accessibility in scientific research.

Following further consultation with our institutional ethics committee, we have now arranged for an official, non-author point of contact to handle all data access requests related to this study:

Data Access Contact:

Beijing Normal University–College of Physical Education and Sports

Ethics Committee Secretariat

Email: ethics.edu@outlook.com

The de-identified data are securely stored in our institution’s research database, under the supervision of the ethics committee. These data will be preserved and made available to qualified researchers upon reasonable request for a minimum of ten years, in accordance with our institutional policies and ethical standards.

Thank you again for your time and continued support.

Sincerely,

Jian Lang

---

## [Decision Letter · Decision Letter 1]

21 Apr 2025

Anthropometric and Strength Characteristics of Adolescent Golfers with Low and High Handicaps

PONE-D-24-29304R1

Dear Dr. Lang,

We’re pleased to inform you that your manuscript has been judged scientifically suitable for publication and will be formally accepted for publication once it meets all outstanding technical requirements.

Kind regards,

Rasool Abedanzadeh, Ph.D

Academic Editor

PLOS ONE

Additional Editor Comments (optional):

Reviewers' comments:

Reviewer's Responses to Questions

**Comments to the Author**

1. If the authors have adequately addressed your comments raised in a previous round of review and you feel that this manuscript is now acceptable for publication, you may indicate that here to bypass the “Comments to the Author” section, enter your conflict of interest statement in the “Confidential to Editor” section, and submit your "Accept" recommendation.

Reviewer #1: All comments have been addressed

Reviewer #2: All comments have been addressed

2. Is the manuscript technically sound, and do the data support the conclusions?

Reviewer #1: Yes

Reviewer #2: Yes

3. Has the statistical analysis been performed appropriately and rigorously? 

Reviewer #1: Yes

Reviewer #2: Yes

4. Have the authors made all data underlying the findings in their manuscript fully available?

Reviewer #1: Yes

Reviewer #2: Yes

5. Is the manuscript presented in an intelligible fashion and written in standard English?

Reviewer #1: Yes

Reviewer #2: Yes

6. Review Comments to the Author

Reviewer #1: The above points have been taken into account correctly. I hope the author can use the mentioned points in his future works.

Reviewer #2: (No Response)

7. PLOS authors have the option to publish the peer review history of their article (what does this mean?). If published, this will include your full peer review and any attached files.

Reviewer #1: No

Reviewer #2: **Yes: **majid mohammadi
